# The Coincidence of Ovarian Endometrioma with Paratubal Leydig Cell Nodules: A Case Report and Literature Review

**DOI:** 10.3390/diagnostics15060703

**Published:** 2025-03-12

**Authors:** Pei-An Chen, Chiu-Hsuan Cheng, Dah-Ching Ding

**Affiliations:** 1Department of Obstetrics and Gynecology, Hualien Tzu Chi Hospital, Buddhist Tzu Chi Medical Foundation, Tzu Chi University, No. 707, Chung-Yang Road, Section 3, Hualien 970, Taiwan; appleball2003@gmail.com; 2Department of Pathology, Hualien Tzu Chi Hospital, Buddhist Tzu Chi Medical Foundation, Tzu Chi University, Hualien 970, Taiwan; chiuhsuan.cheng@gmail.com; 3Institute of Medical Sciences, College of Medicine, Tzu Chi University, Hualien 970, Taiwan

**Keywords:** Leydig cell tumors, endometriosis, ovarian tumor, salpingo-oophorectomy, inhibin

## Abstract

**Background and Clinical Significance:** Paratubal Leydig cell nodules are rare incidental findings that present diagnostic challenges. **Case Presentation**: A 45-year-old female with a history of hypertension and diabetes mellitus presented with fever and chills following an episode of severe dysmenorrhea and menorrhagia. The patient reported heavy menstrual bleeding, persisting for 2–3 years. Physical examination revealed erythema of the perineum and whitish vaginal discharge, with no cervical lesions. Imaging revealed a 15 cm right ovarian cyst. Laboratory investigations showed elevated C-reactive protein (6.37 mg/L) and CA125 (88.82 U/mL) levels, whereas other tumor markers were within normal limits. A pelvic ultrasound revealed a retroverted uterus and a large ovarian mass suggestive of malignancy. The patient underwent a right salpingo-oophorectomy, during which a 15 cm ovarian tumor adherent to the right pelvic sidewall was excised. Histopathological examination revealed an endometriotic cyst with endometrial glandular epithelium positive for estrogen receptor and focal mucinous metaplasia. CD10-positive endometrial stromal cells and paratubal cysts were also observed. Additionally, a small Leydig cell tumor originated from the ovarian hilum was identified and confirmed by positive staining for inhibin, calretinin, and androgen receptors, as well as negative estrogen receptor staining. The postoperative recovery was uneventful, and at the five-week follow-up, the patient’s hormonal levels were normal, and there were no complications. **Conclusions**: This case highlights the importance of thorough histopathological evaluation in managing ovarian masses and the potential coexistence of benign and rare pathological entities, such as Leydig cell tumors.

## 1. Introduction

Paratubal Leydig cell nodules are rare incidental findings that present diagnostic challenges [1]. These nodules, composed of testosterone-producing cells in the ovary and hilum, can occur in extraovarian sites such as the mesovarium or mesosalpinx [1]. While some authors have described them as heterotopic ovarian hilus cells, others argue that their presence in these locations is eutopic. Leydig cells are normally found throughout the ovary, mesovarium, and mesosalpinx [2]. These nodules are often associated with other gynecological abnormalities, such as endometriosis or salpingitis [2]. Paratubal lesions encompass many benign and malignant entities and typically require histological examination for a definitive diagnosis [3]. In rare cases, extrauterine lesions of intermediate trophoblasts can occur in the paratubal region, possibly due to previous ectopic pregnancies [4].

Paratubal Leydig cell nodules are rare testosterone-producing lesions that present diagnostic challenges owing to their small size and inconspicuous nature on imaging [1]. These tumors often manifest with signs of hyperandrogenism, such as hirsutism and voice thickening [5]. Conventional imaging techniques, such as ultrasound, CT, and MRI, may fail to identify these lesions, making diagnosis difficult [6]. In some cases, 18F-FDG PET/CT had successfully localized tumors when other imaging modalities were inconclusive [5]. Expert gynecological ultrasonography remains a crucial diagnostic tool, typically revealing small, solid, and hyperechoic intraovarian nodules [6]. Treatment usually involves unilateral salpingo-oophorectomy, with a generally good prognosis owing to the low malignant potential [7]. Accurate diagnosis and understanding of the prognostic factors are essential for appropriately managing these rare tumors.

Leydig cell tumors are rare ovary neoplasms (<0.5% of ovarian tumors) that typically present with endocrine manifestations like hirsutism [8]. Although usually benign, malignant variants can occur and are characterized by a larger size, infiltrative margins, and higher mitotic rates [8]. Diagnosis is based on morphological features, with Reinke’s crystals being characteristic [8]. Leydig cells can also form nodules at extraovarian sites such as the mesosalpinx, presenting diagnostic challenges owing to their rarity [1]. These paratubal nodules are distinct from ovarian Sertoli-Leydig cell tumors and can exhibit a retiform pattern mimicking other neoplasms [9]. The term “Leydig cell nodule” is preferred over “ovarian hilus cell nest” for extraovarian lesions to avoid confusion [1]. The accurate diagnosis of these entities is crucial for appropriate patient management and prognosis.

Herein, we report a case of a Leydig cell nodule co-occurring with endometrioma.

## 2. Case Report

A 45-year-old female, G1P1, with a history of hypertension and diabetes mellitus, both managed with oral hypoglycemic agents, presented with a fever (38 °C) and chills for one day and subsided thereafter. She reported experiencing unusually severe dysmenorrhea during her menstruation, which began 8 days before admission, along with menorrhagia characterized by the need to change 35 cm pads every hour on the second day of her period. This pattern of heavy bleeding persisted for 2–3 years. The menstrual cycle was regular, and the interval was 28 days, lasting 6 days. The patient denied any history of abdominal surgery. She weighed 78.9 kg, measured 155.3 cm in height, and had a body mass index of 32.7 kg/m^2^. Oily skin and acne were noted.

Upon evaluation in the emergency room, a physical examination revealed an erythematous perineum with whitish vaginal discharge and no cervical lesions. No clitoromegaly was noted. A pap smear was done, and the result showed a reactive change.

Due to a previous fever and chill, laboratory tests with complete blood counts and C-reactive protein (CRP) were performed. It showed a decreased hemoglobin (8.9 g/dL, normal 12–16 g/dL), a normal white blood cell count of 8.92 × 10^3^/μL (normal 3.5–11 × 10^3^/μL), and an elevated CRP of 6.37 mg/L (normal < 1 mg/dL). Tumor markers, including carcinoembryonic antigen (CEA), cancer antigen 125 (CA125), and cancer antigen 19-9 (CA19-9), were 1.64 ng/mL (normal < 3 ng/mL), 88.82 U/mL (normal < 35 U/mL), and less than 0.60 U/mL (normal < 35 U/mL), respectively. The random glucose level was 121 mg/dL (normal < 140 mg/dL).

Computed tomography revealed a 15 cm cyst in the right ovary (Figure 1). No ascites was noted. A pelvic ultrasound revealed a retroverted uterus measuring 6.6 cm × 6.4 cm, an endometrial thickness of 1.3 cm, and a right ovarian tumor of 15 cm. An ovarian malignancy could not be ruled out. Based on these findings, the patient was admitted for a right salpingo-oophorectomy.

The following day, the patient underwent surgery. Prophylactic antibiotics with cefazolin 1 gm and metronidazole 500 mg were prescribed. A large ovarian tumor measuring 15 cm in diameter adherent to the right pelvic sidewall was noted (Figure 2). There were no other intra-abdominal abnormalities. Blood loss was 100 mL, and the operation time was 4 h and 40 min. The procedure proceeded smoothly, and the patient tolerated it well.

The pathology report confirmed the presence of endometriosis with an endometriotic cyst containing estrogen receptor-positive endometrial glandular epithelium and focal mucinous metaplasia. Endometrial stromal cells were CD10-positive. Additionally, paratubal cysts were identified, and a small Leydig cell tumor originated from the ovarian hilum. Histopathology revealed the presence of hilus cells, Reinke crystals, and lipofuscin pigment. Immunohistochemistry revealed positive inhibin, calretinin, androgen receptor staining, and negative estrogen receptor staining (Figure 3).

Postoperatively, the patient recovered uneventfully, with no active bleeding or fever. The surgical wounds were clean and healed well. The patient was discharged in stable condition, with a follow-up scheduled in the outpatient clinic.

At a follow-up visit five weeks after surgery, the patient’s laboratory results, including DHEA-S (135 µg/dL), free T4 (0.81 ng/dL), TSH (0.54 µIU/mL), FSH (7.99 mIU/mL), testosterone (32 ng/dL), and estradiol (30 pg/mL), were all within normal limits. The transvaginal ultrasound showed a uterus measuring 8.6 × 6.6 cm with an endometrial thickness of 4.4 mm. No menorrhagia was noted. The patient remained stable, and no complications were reported.

## 3. Literature Review and Discussion

### 3.1. Search Strategy

A systematic search was conducted using the keywords “Leydig cell tumor, Leydig cell nodule, hilus cell, ovary, or paratubal” from inception to 10 January 2025. Synonyms and related terms were included to expand the scope of this study. Bibliographies of relevant reviews and included studies were also examined. Table 1 provides an overview of the search strategy used in the PubMed, Scopus, Web of Science, and Embase databases.

### 3.2. Definition and Epidemiology

The terminology for these cells is still in debate, with discussions around using “heterotopia” versus “eutopia” [10,11]. Traditionally referred to as ovarian hilus cells (OHCs), some authors advocate for the term ovarian Leydig cells (OLCs) [11,12]. Other terms include hilus cell rest, hilus cell hyperplasia, and hilus cell heterotopia, which reflect varying perspectives on their classification and origin [1,13,14].

Current research investigates the frequency and distribution of ectopic Leydig cells and their relationships with nerve structures [14]. These cells are commonly found in the ovary, fallopian tube, peritoneum, and testes, and theories suggest that their presence is linked to the migration of mesonephric duct cells during embryonic development [11]. Additionally, some studies have identified Leydig cells in specimens obtained after tubal ligations [12].

Leydig cell nodules are clusters of testosterone-producing cells that occur at various locations in the reproductive system. Extragonadal Leydig cell nodules also rarely occur in the mesovarium or mesosalpinx, presenting diagnostic challenges owing to their rarity [1].

Paratubal lesions include Leydig cell nodules, paratubal cysts (PTCs), and endosalpingiosis. PTCs have an incidence of 7.3% in pediatric and adolescent populations, with some cases occurring premenarchally [15]. These nonphysiological cysts may require surgical management for resolution. Using the SEE-Fim protocol, the prevalence of endosalpingiosis, endometriosis, Walthard nests, and PTCs was found to be 22%, 45%, 33%, and 42%, respectively, higher than previously reported [16]. The incidence of most lesions increased with age, except for endometriosis, which decreased after menopause. The clinical significance of endosalpingiosis and its potential association with gynecological malignancies warrants further research. Preoperative diagnosis of paratubal lesions is often limited, and histology is required for a definitive diagnosis [3].

### 3.3. Pathogenesis

The origin and migration of Leydig cells are thought to involve aberrant migration during embryonic development, where ovarian hilus cells (or ovarian Leydig cells) may relocate from the ovarian hilum to ectopic sites such as the fallopian tube or subcapsular ovarian cortex, potentially during Müllerian duct formation [12]. Another theory suggests migration along the mesonephric duct pathway, explaining their frequent presence in the ovaries, fallopian tubes, and peritoneum [17]. In addition, some studies have proposed that undifferentiated stromal cells, perivascular fibroblasts, or perineural fibroblasts may differentiate into Leydig cells under specific stimuli [12].

Hormones play a significant role in the development and proliferation of Leydig cells. Elevated gonadotropin levels, particularly during pregnancy, may stimulate Leydig cell hyperplasia, while these cells produce steroid hormones such as androstenedione, which may further drive their development [11]. In addition, hormonal changes in postmenopausal women can lead to Leydig cell hyperplasia or heterotopia, suggesting that altered hormonal environments contribute to their formation [11].

Leydig cell nodules may be associated with gynecologic conditions such as endometriosis, salpingitis, and ovarian tumors, although it remains unclear whether these conditions are causal or coincidental [12]. These cells are often located near nerve fibers, suggesting a potential relationship between their formation or function, although the mechanism is poorly understood [11]. Additionally, in patients who have undergone chemotherapy, residual tumor cells may sometimes be misidentified as Leydig cell nodules [10].

Paratubal Leydig cell nodules are rare incidental findings in the mesosalpinx and are characterized by testosterone-producing cells with features similar to those of ovarian hilar cells [1]. These nodules are distinct from Leydig cell tumors (LCTs), which are more common in rodents than humans. The lower incidence of LCTs in humans is attributed to the reduced sensitivity of human Leydig cells to proliferative stimuli compared to rats [18]. Although not directly related to paratubal nodules, studies on testicular germ cell tumors have revealed various associated lesions, including Leydig cell hyperplasia, suggesting a potential role for Sertoli cell abnormalities in tumor development [19]. Paratubal lesions encompass many benign and malignant entities, and definitive diagnoses often require histological examination [3].

There is ongoing debate regarding the pathogenesis of Leydig cell nodules, specifically whether they should be classified as “heterotopic” (in an abnormal location) or “eutopic” (in a normal location). Some researchers suggest that if these cells are found in sites such as the ovarian hilum, medulla, cortex, fallopian tube, mesosalpinx, or even within nerve branches, they should be considered “eutopic,” as these may represent normal anatomical locations [14]. The term “heterotopia” should be reserved for instances without anatomical or vascular continuity with the presumed site of origin [14].

### 3.4. Clinical Features

We have summarized the previous case report and case series in Table 2.

Leydig cell nodules are most commonly identified incidentally during routine histopathological examinations of surgical specimens [1]. They are typically small, benign, and not the primary reason for surgical procedures; they are often detected in specimens collected during surgeries for other gynecological conditions [1].

Leydig cell nodules are more commonly found in women aged >40 but can also occur in younger women, indicating that their presence is not limited to any specific age group [1,11].

Leydig cell nodules are commonly located in the mesosalpinx, the fimbrial stroma of the fallopian tube, beneath the ovarian capsule, or within the ovarian cortex away from the ovarian hilum [1,11]. It may also be associated with paratubal cysts or cysts within the ovarian hilum [1,10].

Leydig cell nodules are often associated with various gynecological conditions, including endometriosis, salpingitis isthmica nodosa, ovarian cystic teratomas, endometrial carcinoma, and serous cystadenomas [11]. They can also be detected in patients with histories of primary peritoneal carcinoma or other malignancies [11]. Additionally, there is a potential link between hilar cell heterotopia, hyperplasia, and neoadjuvant chemotherapy (NACT) for ovarian serous carcinoma [10].

Leydig cell nodules can mimic malignancy due to their infiltrative growth pattern, which may resemble metastatic carcinoma, particularly lobular breast carcinoma [11]. Additionally, they can appear similar to perineural or intraneural tumor invasion, potentially leading to misdiagnosis or inappropriate upstaging of malignant tumors [1].

Leydig cell nodules are typically asymptomatic and are most often discovered incidentally during surgery or histopathological examination [1]. They are usually small, often less than 1–1.5 mm in size, although larger nodules are occasionally observed. The nodules do not cause any specific clinical symptoms [1]. The most common symptoms include abdominal pain and menstrual abnormalities (Table 2).

Leydig cell nodules present diagnostic challenges for pathologists due to their rarity and morphological similarities to other cells, particularly at unusual sites [1]. Accurate diagnosis is crucial for differentiating these nodules from metastatic tumors and adrenal rests [12].

Reinke crystals are an inconsistent feature of Leydig cell nodules; while some studies have successfully documented their presence, others have reported an inability to identify them [1].

Leydig cells primarily produce steroid hormones such as androstenedione, with smaller amounts of estradiol and progesterone [11]. Hilus cell hyperplasia, commonly observed in pregnant women, suggests hormonal influences; elevated gonadotropin levels may also contribute to the development and hyperplasia of these cells [12].

In a premenopausal woman with abnormal uterine bleeding and risk factors such as obesity, hypertension, and diabetes mellitus, endometrial cancer should be excluded [25]. Synchronous endometrial and ovarian carcinomas, found in 10% of ovarian and 5% of endometrial cancers, pose a diagnostic challenge in gynecologic pathology [26]. Differentiating independent primary tumors from metastatic disease is crucial for staging and treatment [27]. Histological criteria, including preinvasive lesions and lymphovascular invasion, aid in this distinction, while WT1 and p53 immunostaining help in complex cases [28]. Molecular studies indicate that most of them are clonally related. A study of 22 cases found 12 were metastatic, with 10 true independent primary [29]. Tumors meeting WHO 2020 metastatic criteria had worse outcomes, especially those with p53 abnormalities, while low-grade endometrioid cases with minimal invasion had better prognoses [26]. Prognosis is primarily driven by histologic and molecular features rather than tumor origin.

Imaging is crucial in diagnosing these tumors, which are often small and difficult to localize [30]. The ultrasound characteristics of LCTs include a hypoechoic appearance with marked hypervascularization [31]. On MRI, LCTs typically show low signal intensity on T2-weighted images and marked enhancement on contrast-enhanced sequences [30,32]. Dynamic contrast-enhanced and diffusion-weighted MRI can help differentiate LCTs from normal ovarian stroma [30]. Higher signal intensity on T2-weighted imaging (T2WI) and diffusion-weighted imaging (DWI) and a lower apparent diffusion coefficient (ADC) value were observed on LCT [30]. In cases where conventional imaging fails to identify the tumor, 18F-FDG PET/CT may be useful for localization [5]. The lesion is localized as a focal area of FDG uptake within the tumor [5]. Multimodal imaging approaches provide additional diagnostic information for LCTs, including ultrasound elastography and contrast-enhanced ultrasonography [31].

Immunohistochemistry plays a crucial role in nodule identification and differentiation. Calretinin is a valuable marker for normal and neoplastic Leydig cells, showing strong cytoplasmic and sometimes nuclear staining in LCTs and hyperplasia [33]. INSL3, typically expressed in normal Leydig cells, shows decreased expression in most LCTs but is retained in Leydig cell hyperplasia [34]. Testosterone immunostaining can also assess Leydig cell functionality with variations observed in different patients [35]. Immunohistochemical studies have shown that inhibin is a highly sensitive and specific marker of LCTs with 100% expression in benign and malignant cases [36]. Melan A is also frequently expressed in LCTs, with 86% of positivity reported in steroid cell tumors [37]. Androgen receptor expression is observed in 64% of LCTs [37]. Interestingly, LCTs exhibit a unique estrogen receptor (ER) profile, with strong immunoreactivity for ERβ1 and ERβ2 and detectable ERα expression, which differs from normal Leydig cells that only express ERβ isoforms [38]. Additionally, LCTs show strong aromatase expression, suggesting a potential role of estrogen in tumor development and progression [38]. These findings provide valuable insights into the molecular characteristics of LCTs and contribute to the diagnosis and understanding of their pathogenesis.

### 3.5. Treatment and Management

Most LCTs are discovered incidentally during surgeries for other diseases or cancers. This suggests that LCTs alone may not typically present with conditions requiring surgical intervention. The most common treatments are total hysterectomy and bilateral salpingo-oophorectomy (Table 2).

### 3.6. Prognosis and Follow-Up

Regular follow-up is recommended after surgery for Leydig cell tumors, with the protocol depending on the tumor size, location, and associated symptoms [11]. Given its typically benign nature, the risk of recurrence is low if the tumor is completely excised [1]. In rare cases in which complete excision is not possible, follow-up imaging may be used to monitor any changes.

### 3.7. Future Direction

#### 3.7.1. Pathological Characterization

In cases of endometriosis, careful distinction is needed to differentiate Leydig cell nodules from endometrial tissues with glands and stroma or decidualized stroma using morphology and IHC for confirmation [11].

Molecular diagnostics are not currently specified for Leydig cell tumors; however, future advancements could provide valuable insights into their etiology and classification, potentially enhancing these rare tumors’ diagnostic accuracy and understanding [1].

#### 3.7.2. Clinical Management

Leydig cells produce androgens, primarily testosterone. To guide intervention, a more precise assessment of androgen levels (e.g., testosterone, androstenedione, and DHEA-S) and their correlation with clinical symptoms. Hormonal levels are not routinely measured in cases of small, incidental findings but may be assessed in symptomatic cases or when a larger tumor mass is present. The optimal frequency of hormonal monitoring and imaging to track nodule stability or progression should be defined in the future.

Surgery is the primary treatment for Leydig cell tumors with the goal of complete excision tailored to the tumor’s location and size [10]. Microscopic Leydig cell nodules generally do not require surgical removal unless there is diagnostic uncertainty or concern regarding their malignant potential [1]. Clear guidelines need to be established for when observation is appropriate versus when surgical removal is warranted, particularly in asymptomatic or mildly symptomatic patients.

Microscopic Leydig cell nodules, commonly found incidentally in the mesosalpinx or fallopian tube, are typically benign and do not require treatment [10]. However, careful histopathological evaluation is crucial to differentiate these nodules from other entities, such as ectopic adrenal rests or metastatic carcinoma [12]. Further research is needed to determine whether certain subtypes of Leydig cell nodules have a higher likelihood of transformation and require more aggressive management.

Evaluating the impact of Leydig cell nodules on fertility and ovarian function, guiding fertility preservation strategies when needed. For women wishing to preserve fertility, unilateral salpingo-oophorectomy is usually sufficient if the tumor is confined to one ovary, ensuring minimal impact on reproductive potential [1].

#### 3.7.3. Therapeutic Innovations

Innovation could be explored in non-surgical options, hormonal management, personalized medicine, targeted therapies, immunotherapy, and new technologies such as nanotechnology for targeted drug delivery and imaging, particularly for rare malignant forms of these tumors. Future research will require multicenter studies and registries, interdisciplinary collaboration, and enhanced patient education and awareness.

## 4. Conclusions

Clinical management of Leydig cell tumors primarily involves accurate diagnosis through histopathological examination and IHC. Surgical excision is the main treatment approach if a tumor is present, and the extent of surgery depends on the tumor’s location and size. Monitoring and follow-up are recommended postoperatively, although the risk of recurrence is typically low if the tumor is completely removed. Incidental microscopic Leydig cells are generally benign and do not require treatment; they require differentiation from other similar entities.

## Figures and Tables

**Figure 1 diagnostics-15-00703-f001:**
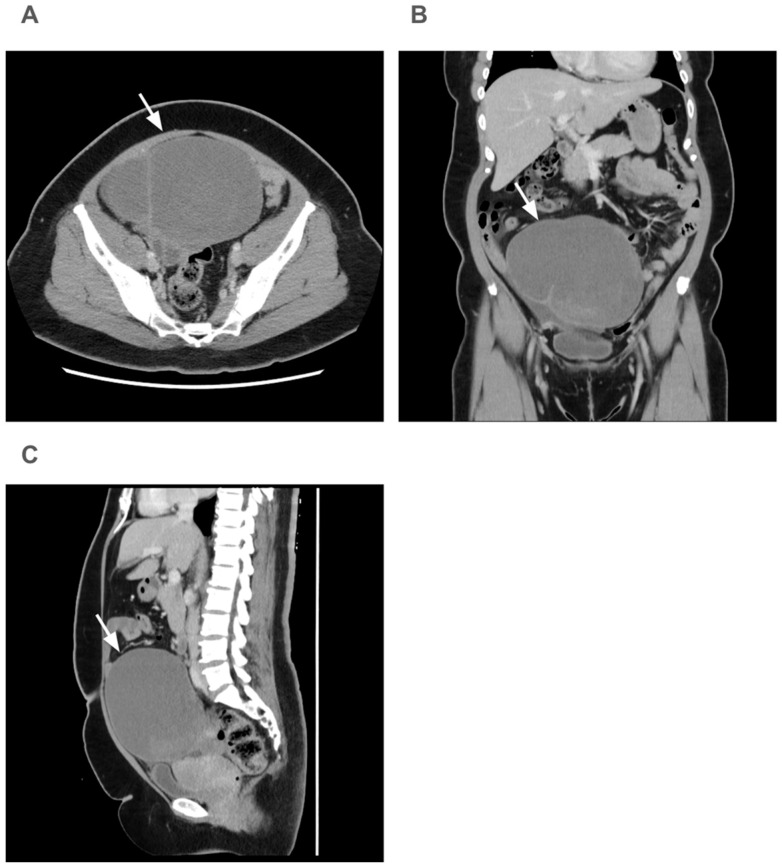
Computer tomography of the ovarian tumor: (**A**) Transverse view (arrow), (**B**) Coronal view (arrow), (**C**) Sagittal view (arrow).

**Figure 2 diagnostics-15-00703-f002:**
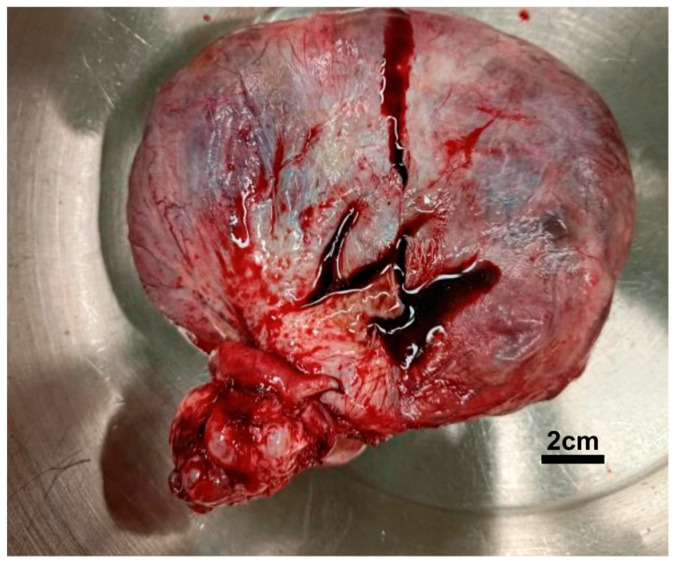
Surgical specimen of ovarian tumor.

**Figure 3 diagnostics-15-00703-f003:**
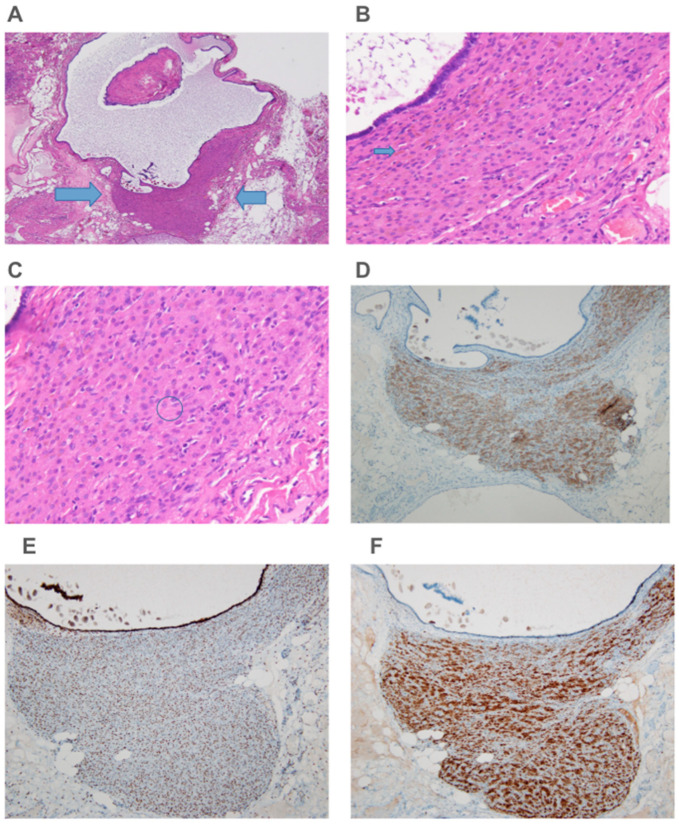
Histology and histochemistry of the Leydig cell nodule. (**A**) Leydig cell nodule (arrow, 40×), (**B**) Lipofuscin pigment (arrow, 400×), (**C**) Reinke crystals (circle, 400×), (**D**) Inhibin (brown color, 100×), (**E**) Androgen receptor (brown color, 100×), (**F**) Calretinin (brown color, 100×).

**Table 1 diagnostics-15-00703-t001:** Search strategy for the literature review.

Items	Specification
Timeframe	From inception to 10 January 2025
Database	PubMed, Scopus, Web of Science, and Embase
Search terms used	“Leydig cell tumor, Leydig cell nodule, hilus cell, ovary or paratubal”
Inclusion and exclusion criteria	All references were SCI-indexed articles written in English.
Selection process	Two independent reviewers evaluated the titles and abstracts to determine eligibility.

**Table 2 diagnostics-15-00703-t002:** Literature review.

Author, Year	Case Age	Symptoms	Lab Finding	Diagnosis	Treatment	Pathological Findings	IHC
USUBÜTÜN et al., 2007 [20]	66	Abdominal pain, vaginal bleeding	CEA, CA153, CA199, CA125: normal	endometrioid adenocarcinoma of endometrium	TAH + BSO + BPLND + PALND + omentectomy	aggregates of hilus cells in the fimbrial part	Inhibin, calretinin, melan a, vimentin
Fischer et al., 2023 [1]	76, 43	n/a	n/a	endometrioid adenocarcinoma of endometrium	TAH + BSO	1.5 mm diameter Leydig cell nodule that was associated with a paratubal cyst	calretinin, alpha-inhibin, and Melan A
Ansari et al., 2014 [21]	52	a mass in the abdomen for two months	n/a	myoma	TAH + BSO	tube had multiple yellowish nodules of varying sizes ranging from 1–3 mm	vimentin, calretinin
Carrasco-Juan et al., 2018 [14]	9 cases	n/a	n/a	myoma, endometrial carcinoma, endometrial hyperplasia, cervical cancer, fallopian tube ligature,	n/a	leydig cells in tube and myoma	n/a
Chougule et al., 2017 [10]	45, 61	abdominal pain and moderate ascites for 1 month	CA125: 2912	serous carcinoma	debulking op	ovarian hilus cells, occasional cells also showed Reinke crystals and lipofuscin pigment	n/a
He et al., 2014 [13]	60	low abdominal pain	CEA, CA199, CA125: normal	serous cystadenoma	BSO	hilus cells in the subepithelial stroma of the fimbrial area	inhibin A and Melan-A
Hirschowitz et al., 2011 [11]	41–78 years, 6 cases	menorrhagia, ovarian pathology	n/a	ovarian endometriosis, a mucinous ovarian tumor, and mature cystic teratoma	n/a	hilus cells were present	n/a
Hu et al., 2020 [12]	63	HSIL of cervix	n/a	HSIL of cervix	LAVH + BSO	hilus cells were present	inhibin, vimentin, and calretinin
Naik et al., 2020 [17]	n/a	ectopic pregnancy	n/a	ectopic pregnancy	salpingectomy	Leydig cells in the tube	n/a
Lewis et al., 1964 [22]	58	abdominal pain, nausea and vomiting	n/a	twisted teratoma	subtotal hysterectomy + bso + appendectomy	ovarian hilus cells, occasional cells showed Reinke crystals and lipofuscin pigment	n/a
Honoré et al., 1979 [23]	24–50, 12 cases	n/a	n/a	n/a	sterilization	hilus cell presented	n/a
Palomaki et al., 1971 [24]	43	abdominal heaviness, weight loss	n/a	myoma	TAH + BSO + appendectomy	hilus cells were present	n/a
Current case, 2025	45	menorrhagia, dysmenorrhea	CA125: 88.82, CEA, CA199: normal	endometriosis	RSO	hilus cells were present, Reinke crystals and lipofuscin pigment+	Inhibin, calretinin, androgen receptor

Footnotes: TAH, total abdominal hysterectomy; BSO, bilateral salpingo-oophorectomy; RSO, right salpingo-oophorectomy; HSIL, high-grade squamous intraepithelial lesion; n/a, not available; IHC, immunohistochemistry; op: operation; BPLND: bilateral pelvic lymph node dissection; PALND: para-aortic lymph node dissection; CEA: carcinoembryonic antigen; LAVH: laparoscopic-assisted vaginal hysterectomy; Lab: laboratory.

## Data Availability

Data is contained within the article.

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
