# Peer review of "The Coincidence of Ovarian Endometrioma with Paratubal Leydig Cell Nodules: A Case Report and Literature Review"

_diagnostics, 2025, doi:10.3390/diagnostics15060703_

Round 1

Reviewer 1 Report

Comments and Suggestions for Authors

From the title of the manuscript presented for review I was expecting a "paratubal" tumor case. Yet the actual description of the case is of a ovarian tumor with mixed components. As such I find the title missleading.

Also, the case is presented very suboptimal: had the patient any clinical signs of hypertestosteron levels? has the vaginal discharge been investigated? preoperative hormonal levels? were there  Reinke’s crystals identified? No explanation was offered for the fever, chills and the abundent bleeding. The entire case presentation needs to be rewritten.

Row 70-71 "Leydig cell tumors are rare neoplasms of the testicular interstitium that typically present with testicular swelling and endocrine manifestations" - please explain the context of testicular neoplasm in a case presentation of a gynecologic tumor in a female patient.

Additional english proofing needs to be done.

I find satisfactory the literature review and that specific part of the manuscript does not need modifications.

The future direction section can do without statements on what is the current standard. It should discuss specific proposed avenues for improving management and diagnostic of such tumors. Please rewrite this section, shorten it and focus on future directions only.

Author Response

Reviewer 1

From the title of the manuscript presented for review I was expecting a "paratubal" tumor case. Yet the actual description of the case is of a ovarian tumor with mixed components. As such I find the title missleading.

Response: We thank the reviewer’s comment. We have changed the title to: “The coincidence of ovarian endometrioma with paratubal Leydig cell nodules: a case report and literature review”. (page 1, line 1)

Also, the case is presented very suboptimal: had the patient any clinical signs of hypertestosteron levels? has the vaginal discharge been investigated? preoperative hormonal levels? were there  Reinke’s crystals identified? No explanation was offered for the fever, chills and the abundent bleeding. The entire case presentation needs to be rewritten.

Response: We thank the reviewer’s comment. We have updated the request information in the history section. No preoperative hormone evaluation was performed due to no speculation of a Leydig cell tumor. The fever and chill subsided in one day. Vaginal discharge was not investigated. BMI was provided. Oily skin and acne were noted for the patient, which may be related to the symptoms of hyperandrogenism. We have rewritten the case presentation. (pages 2-5, lines 82-143) 

Row 70-71 "Leydig cell tumors are rare neoplasms of the testicular interstitium that typically present with testicular swelling and endocrine manifestations" - please explain the context of testicular neoplasm in a case presentation of a gynecologic tumor in a female patient.

Response: We thank the reviewer’s comment. We have replaced the reference with the ovarian Leydig cell tumor. (page 2, lines 68-69)

The statement reads as “Leydig cell tumors are rare ovary neoplasms (<0.5% of ovarian tumors) that typically present with endocrine manifestations like hirsutism [8].”

Additional english proofing needs to be done. 

Response: We thank the reviewer’s comment. We have done English editing by a native speaker. 

I find satisfactory the literature review and that specific part of the manuscript does not need modifications.

Response: We thank the reviewer’s comment. 

The future direction section can do without statements on what is the current standard. It should discuss specific proposed avenues for improving management and diagnostic of such tumors. Please rewrite this section, shorten it and focus on future directions only.

Response: We thank the reviewer’s comment. We have rewritten and shortened the future direction section. (page 11-12, lines 325-363)

The statements read as”3.7 Future direction

3.7.1 Pathological Characterization

In cases of endometriosis, careful distinction is needed to differentiate Leydig cell nodules from endometrial tissue with glands and stroma or decidualized stroma using morphology and IHC for confirmation [11].

Molecular diagnostics are not currently specified for Leydig cell tumors; however, future advancements could provide valuable insights into their etiology and classification, potentially enhancing these rare tumors' diagnostic accuracy and understanding [1].

3.7.2 Clinical Management

Leydig cells produce androgens, primarily testosterone. To guide intervention, a more precise assessment of androgen levels (e.g., testosterone, androstenedione, DHEA-S) and their correlation with clinical symptoms. Hormonal levels are not routinely measured in cases of small, incidental findings but may be assessed in symptomatic cases or when a larger tumor mass is present. The optimal frequency of hormonal monitoring and imaging to track nodule stability or progression should be defined in the future.

Surgery is the primary treatment for Leydig cell tumors with the goal of complete excision tailored to the tumor's location and size [10]. Microscopic Leydig cell nodules generally do not require surgical removal unless there is diagnostic uncertainty or concern regarding their malignant potential [1]. Establishing clear guidelines for when observation is appropriate versus when surgical removal is warranted, particularly in asymptomatic or mildly symptomatic patients.

Microscopic Leydig cell nodules, commonly found incidentally in the mesosalpinx or fallopian tube, are typically benign and do not require treatment [10]. However, careful histopathological evaluation is crucial to differentiate these nodules from other entities, such as ectopic adrenal rests or metastatic carcinoma [12]. Further research is needed to determine whether certain subtypes of Leydig cell nodules have a higher likelihood of transformation and require more aggressive management.

Evaluating the impact of Leydig cell nodules on fertility and ovarian function, guiding fertility preservation strategies when needed. For women wishing to preserve fertility, unilateral salpingo-oophorectomy is usually sufficient if the tumor is confined to one ovary, ensuring minimal impact on reproductive potential [1].

3.7.3 Therapeutic Innovations

Innovation could be explored in non-surgical options, hormonal management, personalized medicine, targeted therapies, immunotherapy, and new technologies such as nanotechnology for targeted drug delivery and imaging, particularly for rare malignant forms of these tumors. Future research will require multicenter studies and registries, interdisciplinary collaboration, and enhanced patient education and awareness.”

Reviewer 2 Report

Comments and Suggestions for Authors

Thank you for interesting paper. I have some comments:

-In a premenopausal woman with abnormal uterine bleeding and risk factors such as obesity, hypertension and diabetes mellitus, the endometrial cancer should be excluded. Please refer to this document doi: 10.4103/jmh.jmh_81_22.

-The authors should expand the discussion concerning Synchronous Endometrial and Ovarian Carcinomas. doi: 10.1097/PGP.0000000000000982. 

-Indicated markers should added to Fig 1.

-Full-words should be provided under table 2 as foot-note.

-Obstetric history (gravida, parity) should be given.

-BMI should be added.

-Abdominal circumference should be described in case of large abdominal tumor.

-Abdominal fluid existed or not? 

Comments on the Quality of English Language

English editing is required.

Author Response

Reviewer 2

Thank you for interesting paper. I have some comments:

-In a premenopausal woman with abnormal uterine bleeding and risk factors such as obesity, hypertension and diabetes mellitus, the endometrial cancer should be excluded. Please refer to this document doi: 10.4103/jmh.jmh_81_22.

Response: We thank the reviewer’s comment. We have added the suggested reference and content in the discussion section. (page 10, line 270-271)

The statements read as”In a premenopausal woman with abnormal uterine bleeding and risk factors such as obesity, hypertension, and diabetes mellitus, endometrial cancer should be excluded [25].”

-The authors should expand the discussion concerning Synchronous Endometrial and Ovarian Carcinomas. doi: 10.1097/PGP.0000000000000982. 

Response: We thank the reviewer’s comment. In the discussion section, We added a paragraph regarding synchronous endometrial and ovarian carcinomas.(page 10, lines 272-282)

The statements read as”Synchronous endometrial and ovarian carcinomas, found in 10% of ovarian and 5% of endometrial cancers, pose a diagnostic challenge in gynecologic pathology [26]. Differentiating independent primary tumors from metastatic disease is crucial for staging and treatment [27]. Histological criteria, including preinvasive lesions and lymphovascular invasion, aid in this distinction, while WT1 and p53 immunostaining help in complex cases [28]. Molecular studies indicate most are clonally related. A study of 22 cases found 12 were metastatic, with 10 true independent primary [29]. Tumors meeting WHO 2020 metastatic criteria had worse outcomes, especially those with p53 abnormalities, while low-grade endometrioid cases with minimal invasion had better prognoses [26]. Prognosis is primarily driven by histologic and molecular features rather than tumor origin.”

-Indicated markers should added to Fig 1. 

Response: We thank the reviewer’s comment. We have added an arrow to point out the ovarian tumor. (Figure 1, page 3)

-Full-words should be provided under table 2 as foot-note.

Response: We thank the reviewer’s comment. We have added full-word spelling in the footnote of Table 2. (page 9, lines 226-230)

The statements read as”Footnotes: TAH, total abdominal hysterectomy; BSO, bilateral salpingo-oophorectomy; RSO, right salpingo-oophorectomy; HSIL, high-grade squamous intraepithelial lesion; n/a, not available; IHC, immunohistochemistry; op: operation; BPLND: bilateral pelvic lymph node dissection; PALND: para-aortic lymph node dissection; CEA: carcinoembryonic antigen; LAVH: laparoscopic-assisted vaginal hysterectomy; Lab: laboratory.”

-Obstetric history (gravida, parity) should be given.

Response: We thank the reviewer’s comment. We have added G1P1 to the history. (page 2, line 82) 

-BMI should be added.

Response: We thank the reviewer’s comment. We have added BMI in the history (page 2, lines 89-90) 

The statements read as “She weighed 78.9 kg, measured 155.3 cm in height, and had a body mass index of 32.7 kg/m².”

-Abdominal circumference should be described in case of large abdominal tumor.

Response: We thank the reviewer’s comment.  We did not measure the abdominal circumference. 

-Abdominal fluid existed or not? 

Response: We thank the reviewer’s comment. No ascites was noted.  (page 3, lines 103-104)

Round 2

Reviewer 1 Report

Comments and Suggestions for Authors

The authors have addressed all my concerns adequately and I believe the manuscript can be published in the current form. 

Reviewer 2 Report

Comments and Suggestions for Authors

Thank you for revision.

The paper is well-improved.